# Distribution pattern, molecular transmission networks, and phylodynamic of hepatitis C virus in China

**Jingrong Ye[1◐], Yanming Sun[1◐], Jia Li[1◐], Xinli Lu[2◐], Minna Zheng[3◐], Lifeng Liu[4], Fengting Yu[5], Shufang He[1], Conghui Xu[1], Xianlong Ren[1], Juan Wang[1], Jing Chen[1], Yuhua Ruan[6], Yi Feng[6], Yiming Shao[6◐]\*, Hui Xing[6◐]\*, Hongyan Lu[1◐]\***

**1** Institute for HIV/AIDS and STD Prevention and Control, Beijing Center for Disease Prevention and Control (CDC), Dong Chen District, Beijing, China, **2** Institute for HIV/AIDS and STD Prevention and Control, Hebei CDC, Shijiazhuang, Hebei, China, **3** Institute for HIV/AIDS and STD Prevention and Control, Tianjin CDC, Hedong District, Tianjin, China, **4** Center for Infectious Diseases, Beijing YouAn Hospital, Capital Medical University, Feng Tai District, Beijing, China, **5** Clinical and Research Center of Infectious Diseases, Beijing Ditan Hospital, Capital Medical University, Chaoyang District, Beijing, China, **6** Division of Virology and Immunology, State Key Laboratory of Infectious Disease Prevention and Control (SKLID), Collaborative Innovation Center for Diagnosis and Treatment of Infectious Diseases, National Center for AIDS/STD Prevention and Control (NCAIDS), China CDC, Changping District, Beijing, China

◐ These authors contributed equally to this work.
\* hongyanlu1972@sina.com (HL); xingh@chinaaids.cn (HX); yshao@bjmu.edu.cn (YS)

**Data Availability Statement:** The minimal anonymized data sets necessary to replicate study findings have been uploaded in Open Science Framework (OSF) DOI: https://doi.org/10.17605/

## Abstract

In China, few molecular epidemiological data on hepatitis C virus (HCV) are available and all previous studies were limited by small sample sizes or specific population characteristics. Here, we report characterization of the epidemic history and transmission dynamics of HCV strains in China. We included HCV sequences of individuals belonging to three HCV surveillance programs: 1) patients diagnosed with HIV infection at the Beijing HIV laboratory network, most of whom were people who inject drugs and former paid blood donors, 2) men who have sex with men, and 3) the general population. We also used publicly available HCV sequences sampled in China in our study. In total, we obtained 1,603 *Ns5b* and 865 *C/E2* sequences from 1,811 individuals. The most common HCV strains were subtypes 1b (29.1%), 3b (25.5%) and 3a (15.1%). In transmission network analysis, factors independently associated with clustering included the region (OR: 0.37, 95% CI: 0.19–0.71), infection subtype (OR: 0.23, 95% CI: 0.1–0.52), and sampling period (OR: 0.43, 95% CI: 0.27–0.68). The history of the major HCV subtypes was complex, which coincided with some important sociomedical events in China. Of note, five of eight HCV subtype (1a, 1b, 2a, 3a, and 3b), which constituted 81.8% HCV strains genotyped in our study, showed a tendency towards decline in the effective population size during the past decade until present, which is a good omen for the goal of eliminating HCV by 2030 in China.

## Introduction

Hepatitis C virus (HCV) infection is a major public health threat in China. The most recent estimate of the national prevalence of HCV infection is 0.7%, representing approximately 10

OSF.IO/NKD8Y. HCV sequences have been submitted to GenBank (See the Supporting Information file for accession numbers).

**Funding:** This work was supported by China Capital's Funds for Health Improvement and Research (2022-1G-3011) to Jingrong Ye, Beijing Municipal Science & Technology Commission (D161100000416002), Beijing High-Level Public Health Doctor Cultivation Project (Academic Leader-01-04) to Hongyan Lu, Cultivation Fund of Beijing Center for Disease Prevention and Control (2019-BJYJ-13) to Yanming Sun. The funders had no role in study design, data collection and analysis, decision to publish, or preparation of the manuscript.

**Competing interests:** The authors have declared that no competing interests exist.

million people [1]. In 2020 alone, 194,066 individuals were newly diagnosed with HCV infection [2]. Moreover, an estimated 34,198 people died of cirrhosis attributed to HCV infection in 2017 [3]. Inspired by the exciting curable therapeutic effect of new all-oral antivirals with a short treatment duration, more manageable side effects, and improved sustained virologic response (SVR), in 2016, the WHO introduced ambitious global targets to eliminate HCV infection by 2030 [4,5]. To achieve these goals, China needs to develop national policies based on up-to-date and reliable epidemiological data. Previous national or quasi-national studies have determined HCV genotype distribution in China. However, all these studies are limited by small sample sizes, samples of a specific population (mostly from people who inject drugs [PWID] and former paid blood donors, whereas men who have sex with men [MSM] were seldom considered) and restricted geographic sampling [6–11]. Previous studies have also reconstructed the evolutionary history of HCV lineages in China and successfully linked the time scale of HCV evolution to unique historical events and past sociomedical conditions in China, such as the "Cultural Revolution" and "Encouraged Plasma Campaign" [12,13]. However, the outcomes of these theoretical studies have been limited by a relatively narrow span of sampling time.

Rapidly evolving RNA viruses, such as HIV and HCV, contain measurable footprints in their genome, which can be used for molecular transmission networks. Thus, by using nucleotide sequences, HIV transmission networks that link people who are infected with genetically similar isolates can be constructed, whereby linked people are presumed to have a direct or indirect epidemiologic connection and usually represent a "hotspot" of HIV transmission [14–16]. Over the last two decades, many clustering methods have been developed to define HIV transmission networks within a population. Broadly speaking, these methods can be grouped into two categories: methods that cluster directly on sequence variation via pairwise genetic distance measures, and methods that interpret this variation in the context of subtrees in a phylogeny. Phylogenetic analysis can be associated with high computational burden, especially for large sequence datasets. However, the genetic distance method can be computed rapidly. Therefore recent network analyses have favoured the generally faster and parameter-rich distanced-based methods [17,18].

These network analyses contribute significantly to our understanding of HIV epidemiology, for example, by providing information about HIV epidemics by identifying transmission linkages and by elucidating differences in transmission within and between populations [14–16]. HCV also evolves rapidly and shares the same routes of transmission as HIV; however, HCV transmission networks have never been characterized in China.

In this study, we aimed to update the genotype distribution, infer the molecular transmission networks and reconstruct the epidemic history of HCV in China using a substantially more comprehensive dataset and metadata than previous works.

## Methods

### Sampling strategy

We designed a cross-sectional study to make full use of all available HCV genotyping data in China. The study population consisted of four separate groups of HCV-infected individuals (Fig 1 and S1 Table). The first group consisted of patients diagnosed with HIV infection at the Beijing HIV laboratory network (BHLN) from 1999 through 2017. The BHLN, established in 1986, is a collaborative network of laboratories involved in HIV diagnosis. It was authorized by the Beijing Municipal Commission of Health and includes a central reference laboratory in the Beijing Center for Disease Prevention and Control (CDC), four additional HIV reference

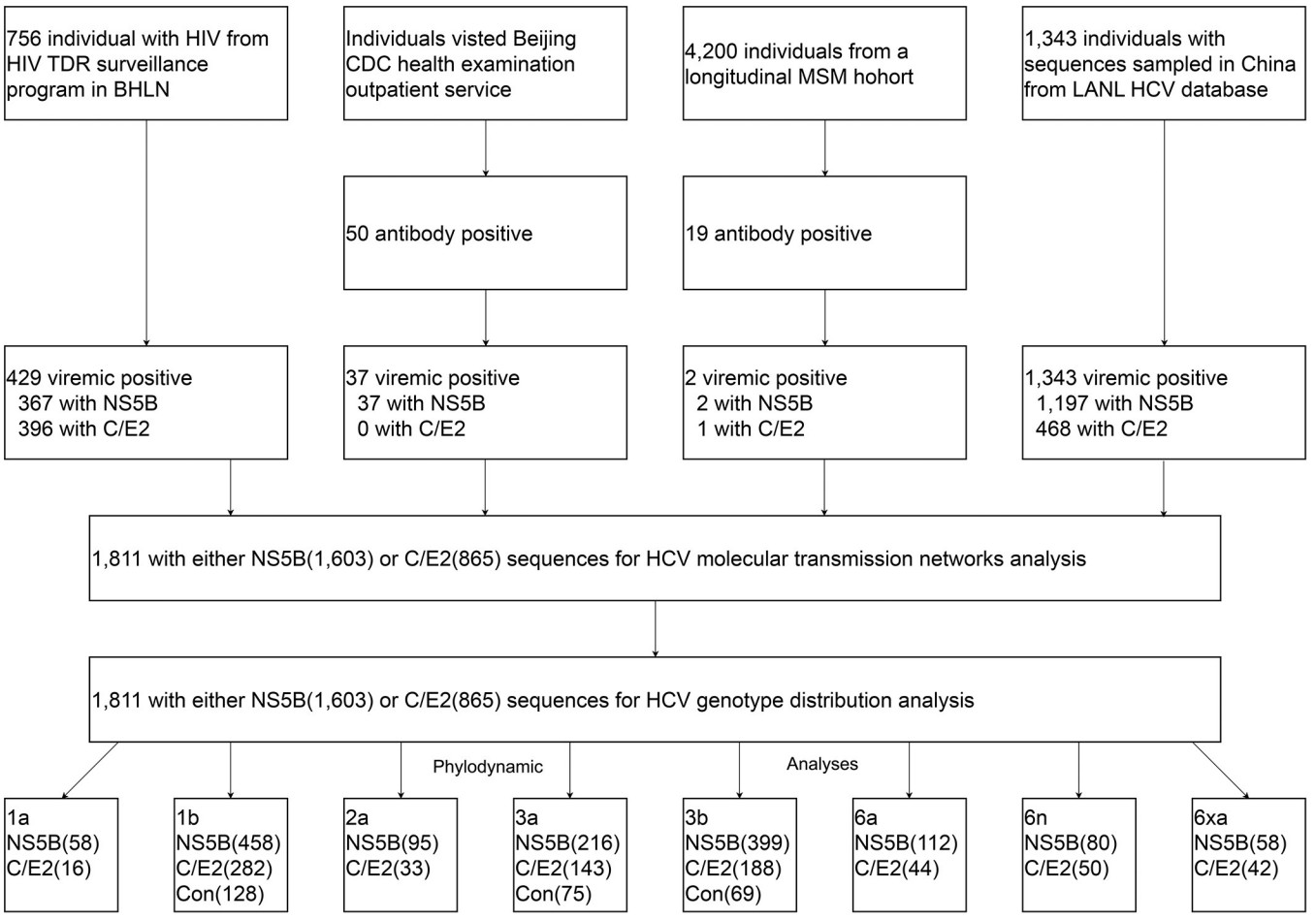

**Fig 1. Study profile.** BHLN = Beijing HIV laboratory network. LANL = Los Alamos National Laboratory. TDR = Transmitted drug resistance. MSM = Men who have sex with men. PWID = People who inject drugs.

laboratories (DiTan, YouAn, Peking Union Medical College, and People's Liberation Army of China [PLA] General Hospital), and approximately 280 HIV screening laboratories [19,20].

To ensure that as many sequences as possible were obtained, we adopted a cost-effective sampling strategy, which is one that obtains many sequences at a low cost. We mainly considered the two groups of populations most seriously affected by HCV infection in the BHLN: PWID and former paid blood donors. We acknowledge that we obtained HCV RNA first and designed a sampling strategy later. We obtained RNA remnants from China's national HIV transmitted drug resistance surveillance programme conducted by the BHLN. This programme randomly selected approximately 40% of individuals with newly diagnosed HIV infection between 1999 and 2017 from the national HIV epidemic database maintained by the BHLN. In total, we obtained 9,059 RNA samples from heterosexual individuals (2,190), MSM (6,136), PWID (539), and former paid blood donors (194) [19,20]. Unfortunately, for most of these samples, HCV antibody test records were not available (except for four heterosexuals and two volunteer blood donors). Therefore, we devised cost-effective inclusion criteria. According to the literature, the national prevalence of HCV co-infection in people living with HIV is approximately 4.0%, 6.4%, 82.4%, and 92.9% for heterosexuals, MSM, PWID, and former paid blood donors [6]. There is an unduly high HCV prevalence among the PWID and

former paid blood donors. In other words, if we were concerned with these two groups of populations, we could obtain more HCV sequences at a lower cost. Moreover, or most importantly, the plasma samples were not sufficient for performing another HCV antibody test. Therefore, we included all samples from PWID and former paid blood donors in the BHLN in our analysis. Four heterosexual and two volunteer blood donors who were HCV antibody positive were also included.

The second group consisted of individuals who visited the health examination outpatient service of the Beijing CDC in 1999 and had HCV antibody-positive records. We roughly deemed these individuals to be from the general population.

The third group consisted of individuals with HCV antibody-positive records from an MSM cohort. In this cohort, we conducted seven serial consecutive cross-sectional surveys of MSM from 2015 through 2021. The purpose of this survey was to track trends in the prevalence of HIV, HCV and syphilis in this population [21,22].

The fourth group consisted of publicly available sequences from the Los Alamos National Laboratory (LANL) HCV sequence database. We retrieved all sequences sampled in China with information on the province of origin and sample year and covering the same genomic region from the databases (data available as of Dec 1, 2021). The sampling year and locations were confirmed by reference to the original literature.

## Patient inclusion and data collection

We extracted baseline data on individuals from the national HIV epidemic database or LANL HCV database, including demographic and population characteristics and CD4 cell count. For geographic location, we grouped individuals into 25 provinces according to Hukou, a basic system of household registration in China. It officially identifies a person as a resident of an area and includes identifying information such as name, parents, spouse, and date of birth.

## Genotypic analysis

We performed population-based sequencing of the HCV *Ns5b* and *C/E2* regions in all specimens using in-house methods [8]. These sequences correspond to nucleotides 8,400 to 9,100 and 927 to 2,040, respectively, in the H77 genome. We inferred HCV genotype by automated genotyping in context-based modelling for expeditious typing (COMET)-HCV [23], followed by maximum likelihood (ML) phylogenetic analysis of the sequences [24]. An ML tree was used to confirm the results of COMET.

## Transmission network analysis

To construct the transmission network, we followed the protocol outlined by Wertheim et al. [14–16]. We aligned HCV sequences in a pairwise fashion and then evaluated Tamura-Nei 93 (TN93) distances for all sequences using the HyPhy package [25]. TN93 genetic distance was used because it can be computed rapidly and is the most complex genetic distances that can be represented by a closed-form solution [17,18].We performed stepwise transmission network analysis using a serial set of genetic threshold (0.005–0.045 subsitution/site, increment every 0.0025 subsitution/site) [26]. We selected 0.01 subsitution/site and 0.0325 substituion/site for *Ns5b* and *C/E2* datasets, because this distance identifies the maximum number of clusters in the transmission network (S1 Fig). The degree (connectivity) of each individual was defined as the number of links (edges in the transmission network) to other individuals. Clusters were defined as connected components of the network comprising two or more nodes. We used Cytoscape (3.8.0) to visualize the networks.

## Phylogenetic analysis

We aligned sequences by using the BioEdit tool and manually corrected the alignment according to the encoded reading frame. If several sequences from the same patient were available in the dataset, we retained only the oldest sequence. Long branch trees were reconfirmed regarding genotype, and those found to be misclassified were eliminated. All these efforts help to minimize the possibility of duplicate patient sampling. We reconstructed ML phylogenetic tree with the datasets using the GTR+CAT nucleotide substitution model in FastTree 2.1 [24]. Temporal signal was examined using root-to-tip regression in TempEst v1.5.3 [27]. The sequences whose sampling year is incongruent with genetic divergence were excluded for Bayesian analysis. We estimated time-calibrated phylogenies dated from time-stamped genome data using the Bayesian software package BEAST(version 1.10.4) [28]. We only did Bayesian evolutionary analysis for main eight HCV subtype (1a, 1b, 2a, 3a, 3b, 6a, 6n, and 6xa) because their datasets contain at least 10 dated sequences. We used the HKY nucleotide substitution model with codon partitions [29] and Bayesian SkyGrid tree prior [30] with an uncorrelated relaxed clock with a lognormal distribution [31,32].

For each dataset, at least three independent Markov chain Monte Carlo (MCMC) chains were run for 50 million generations with states sampled every 1,000 generations. Multiple MCMC chains were calculated to increase Effective Sample Size(ESS). Log files were combined using Logcombiner (v.1.10.4) to ensure sufficient convergence (ESS≧200) with 10% of posterior samples discarded as burn-in. MCMC mixing was diagnosed using visual trace inspection and calculation of ESS in Tracer (v.1.7.2) [33]. The ESS of a parameter sampled from an MCMC is the number of effectively independent draws from the posterior distribution that the Markov chain is equivalent to. Maximum clade credibility trees were summarized using TreeAnnotator after discarding 10% as burn-in (S1 File.The protocol for Bayesian estimation of past population dynamics using the Skygrid coalescent model.).

## Ethical issues

All analyses were performed on de-identified datasets to protect the participants' anonymity. The research ethics committee at the Beijing CDC approved this study, and all the methods in this study were performed in accordance with approved guidelines. By law (Law of the People's Republic of China on the Prevention and Treatment of Infectious Diseases, and Regulations on AIDS Prevention and Treatment), consent was not required, as these data were collected and analysed in the course of routine public health surveillance.

## Statistical analysis

Four sampling phases were established: 1994–2003, 2004–2008, 2009–2013, and 2014–2020. The most early (1994–2003) and recent (2014–2020) phases encompassed more years to account for the relatively fewer data available in these years. We compared categorical data with the x2 test and continuous data with one-way ANOVA, wherever appropriate. We analysed the variables for clustering using univariable and multivariable logistic regression. Variables considered were region, HCV subtype, population characteristics, and sampling phase. We analysed all variables separately and entered those associated ($P<0.1$) with the outcomes into the multivariable model. We present the results as odds ratios (ORs) with 95% confidence intervals (CIs). We performed all analyses using R (version 4.1.1; R Foundation, Vienna, Austria). We used listwise deletion to handle missing data.

# Results

## Study population

Our study population was four cohorts (Fig 1 and S1 Table). First, we included 756 individuals newly diagnosed with HIV from the national epidemiology database of China. The BHLN is authorized officially to participate in maintaining this database. Second, we included 50 individuals who visited the health examination outpatient service of the Beijing Center for Disease Prevention and Control (CDC) and had HCV antibody-positive records. Third, we included 19 individuals with HCV antibody-positive records from an MSM cohort that consist of 4,200 people recruited between 2015 and 2021.

From the above three cohorts, we included 825 individuals in our analysis. Amplification and sequencing of *Ns5b* and *C/E2* fragments were successful for 342 (40.6%) individuals. For an additional 126 patients, sequences were obtained for either *Ns5b* alone (n = 64) or *C/E2* fragment alone (n = 62). Hence, we were able to perform HCV genotyping and phylogenetic analysis for 468 (56.7%) individuals based on the availability of sequence data. The prevalence of viraemic HCV infection in PWID, former paid blood donors, and MSM was 60.7% (327 of 539), 40.7% (79 of 194), and 0.05% (2 of 4,200) respectively. The majority of the participants were men (80.1%). Han, Uygur and Yi ethnicities accounted for 43.0%, 38.4% and 10.3%, respectively. The median age was 32 years (interquartile range [IQR] 26–39). The CD4 was only available for individuals with HIV/HCV co-infection and the overall median baseline CD4 count was 336 cells per μL (IQR 240–461).

Fourth, we included all *Ns5b* and *C/E2* sequences sampled in China with known sampling provinces and sampling years available in the LANL HCV sequence database. After rigorous phylogenetic analysis, we obtained both HCV *Ns5b* and *C/E2* sequence fragments from 322 individuals and either of the fragments from 1,021(S2 File.Accession numbers).

Thus, we included 1,603 *Ns5b* and 865 *C/E2* sequences from 1,811 individuals from 25 provinces of China in the final analysis (Fig 1 and S1 Table). The transmission risk group were predominantly PWID (77.1%), followed by general population (16.5%), former paid blood donor (5.7%), heterosexual (0.3%), MSM(0.1%), and volunteer blood donor (0.1%) (Table 1).

## Phylogenetic analysis

We performed phylogenetic analysis using the merged *Ns5b* and *C/E2* sequence dataset, which consisted of 1,603 and 865 sequences respectively. The phylogenetic tree confirmed the genotype assignment by COMET-HCV, and the genotype determinations between the *Ns5b* and *C/E2* fragments were consistent (S2 Fig). All isolates in our study belong to four genotypes (1, 2, 3, and 6) and 13 subtypes (1a, 1b, 2a, 3a, 3b, 6a, 6e, 6g, 6l, 6n, 6v, 6w, and 6xa). The prevalence of genotypes 1, 2, 3, and 6 was 32.6%, 6.7%, 40.6%, and 20.1%, respectively. The most common HCV subtypes in order of decreasing frequency were 1b (29.1%), 3b (25.5%), 3a (15.1%), 6a (7.5%), 2a (6.7%), 6n (6.6%), 6xa (3.8%), and 1a (3.5%). Additional clades, including subtypes 6e, 6g, 6l, 6w, and 6v, were present in fewer than 1.0% of individuals. HCV genotype patterns differed between population groups. In most groups, subtype 1b was the most prevalent (Tables 1 and 2). Table 1 presents the temporal trends for these eight major subtypes. There was a decreasing trend for genotype 1b and a stable trend for 3a and 3b. Table 2 illustrates the geographical distribution of HCV subtypes in China.

## Network inference

Using the *Ns5b* sequence (n = 1,603), we built an HCV transmission network representing 25 provinces of China. The network contains 111 connected components with ≥2 nodes

**Table 1. Baseline characteristic by sampling phase.**

|  | 1994–2003 | 2004–2008 | 2009–2013 | 2014–2020 | Total |
|---|---|---|---|---|---|
| Sex |  |  |  |  |  |
| Men | 123(85.4) | 271(80.7) | 69(78.4) | 48(68.6) | 511(80.1) |
| Women | 21(14.6) | 65(19.3) | 19(21.6) | 22(31.4) | 127(19.9) |
| Age at diagnosis(years)[a] | 27(23–31) | 32(26–37) | 34(28–39) | 44(33–72) | 32(26–39) |
| CD4 counts(cells per μL)[b] | 213(119–306) | 294(206–422) | 369(286–506) | 356(299–405) | 336(240–461) |
| Ethnicity |  |  |  |  |  |
| Han | 71(75.5) | 86(36.1) | 47(53.4) | 9(12) | 213(43) |
| Uighur | 23(24.5) | 124(52.1) | 25(28.4) | 18(24) | 190(38.4) |
| Yi | 0(0) | 22(9.2) | 16(18.2) | 13(17.3) | 51(10.3) |
| Li | 0(0) | 0(0) | 0(0) | 34(45.3) | 34(6.9) |
| Other Minority | 0(0) | 6(2.5) | 0(0) | 1(1.3) | 7(1.4) |
| Population characteristic[c] |  |  |  |  |  |
| Heterosexual | 2(1) | 0(0) | 2(0.3) | 0(0) | 4(0.3) |
| MSM | 0(0) | 0(0) | 0(0) | 2(0.7) | 2(0.1) |
| PWID | 153(77.3) | 180(75.9) | 442(67.7) | 288(99.3) | 1063(77.1) |
| Former paid blood donor | 4(2) | 57(24.1) | 18(2.8) | 0(0) | 79(5.7) |
| General population | 37(18.7) | 0(0) | 191(29.2) | 0(0) | 228(16.5) |
| Volunteer blood donor | 2(1) | 0(0) | 0(0) | 0(0) | 2(0.1) |
| Region |  |  |  |  |  |
| North | 13(6) | 40(11.5) | 18(2) | 2(0.6) | 73(4) |
| Northeast | 2(0.9) | 7(2) | 2(0.2) | 0(0) | 11(0.6) |
| East | 0(0) | 5(1.4) | 367(39.8) | 0(0) | 372(20.5) |
| Central South | 2(0.9) | 37(10.7) | 281(30.4) | 43(13.3) | 363(20) |
| Southwest | 96(44.2) | 128(36.9) | 226(24.5) | 257(79.3) | 707(39) |
| Northwest | 54(24.9) | 128(36.9) | 27(2.9) | 20(6.2) | 229(12.6) |
| Unknown | 50(23) | 2(0.6) | 2(0.2) | 2(0.6) | 56(3.1) |
| Genotype and subtype |  |  |  |  |  |
| 1a | 2(0.9) | 2(0.6) | 38(4.1) | 22(6.8) | 64(3.5) |
| 1b | 80(36.9) | 121(34.9) | 291(31.5) | 35(10.8) | 527(29.1) |
| 2a | 18(8.3) | 18(5.2) | 85(9.2) | 0(0) | 121(6.7) |
| 3a | 43(19.8) | 62(17.9) | 104(11.3) | 64(19.8) | 273(15.1) |
| 3b | 48(22.1) | 99(28.5) | 213(23.1) | 102(31.5) | 462(25.5) |
| 6a | 8(3.7) | 10(2.9) | 91(9.9) | 26(8) | 135(7.5) |
| 6n | 11(5.1) | 18(5.2) | 50(5.4) | 40(12.3) | 119(6.6) |
| 6xa | 7(3.2) | 12(3.5) | 36(3.9) | 13(4) | 68(3.8) |
| Other | 0(0) | 5(1.4) | 15(1.6) | 22(6.8) | 42(2.3) |

Data are n (%)

North = Beijing, Hebei, Shanxi, Inner Mongolia

Northeast = Liaoning, Heilongjiang

East = Shanghai, Jiangsu, Zhejiang, Anhui, Jiangxi, Shandong

Central South = Henan, Hubei, Hunan, Guangdong,Guangxi, Hainan

Southwest = Chongqing, Sichuan, Guizhou, Yunnan

Northwest = Shannxi, Qinghai, Sinkiang

[a] Data for n = 429

[b] Data for n = 125

[c]Data for n = 1378

[d]Data for n = 1755

MSM = Men who have sex with men

PWID = People who inject drugs

Other = 6e, 6 g, 6 l, 6w, and 6v.

**Table 2. HCV genotype and subtype assignment by selected characteristics.**

| | 1a | 1b | 2a | 3a | 3b | 6a | 6n | 6xa | Other | Total |
|---|---|---|---|---|---|---|---|---|---|---|
| Sex[a] | | | | | | | | | | |
| Men | 7(1.4) | 162(31.7) | 17(3.3) | 106(20.7) | 140(27.4) | 11(2.2) | 26(5.1) | 29(5.7) | 13(2.5) | 511(100) |
| Women | 1(0.8) | 45(35.4) | 11(8.7) | 17(13.4) | 22(17.3) | 15(11.8) | 3(2.4) | 1(0.8) | 12(9.4) | 127(100) |
| Ethnicity[b] | | | | | | | | | | |
| Han | 3(1.4) | 101(47.4) | 35(16.4) | 21(9.9) | 38(17.8) | 11(5.2) | 2(0.9) | 1(0.5) | 1(0.5) | 213(100) |
| Uyghur | 2(1.1) | 89(46.8) | 0(0) | 57(30.0) | 40(21.1) | 2(1.1) | 0(0) | 0(0) | 0(0) | 190(100) |
| Yi | 1(2) | 3(5.9) | 0(0) | 14(27.5) | 11(21.6) | 0(0) | 0(0) | 22(43.1) | 0(0) | 51(100) |
| Li | 0(0) | 0(0) | 0(0) | 0(0) | 0(0) | 12(35.3) | 0(0) | 0(0) | 22(64.7) | 34(100) |
| Other minority | 0(0) | 3(42.9) | 0(0) | 2(28.6) | 2(28.6) | 0(0) | 0(0) | 0(0) | 0(0) | 7(100) |
| Population characteristic[c] | | | | | | | | | | |
| Heterosexual | 0(0) | 3(75) | 1(25) | 0(0) | 0(0) | 0(0) | 0(0) | 0(0) | 0(0) | 4(100) |
| MSM | 0(0) | 2(100) | 0(0) | 0(0) | 0(0) | 0(0) | 0(0) | 0(0) | 0(0) | 2(100) |
| PWID | 49(4.6) | 202(19) | 6(0.6) | 241(22.7) | 339(31.9) | 58(5.5) | 96(9) | 65(6.1) | 7(0.7) | 1063(100) |
| Former paid blood donor | 0(0) | 52(65.8) | 16(20.3) | 3(3.8) | 6(7.6) | 1(1.3) | 1(1.3) | 0(0) | 0(0) | 79(100) |
| General population | 0(0) | 169(74.1) | 37(16.2) | 5(2.2) | 8(3.5) | 7(3.1) | 2(0.9) | 0(0) | 0(0) | 228(100) |
| Volunteer blood donor | 0(0) | 1(50) | 0(0) | 0(0) | 1(50) | 0(0) | 0(0) | 0(0) | 0(0) | 2(100) |
| Region[d] | | | | | | | | | | |
| North | 0(0) | 33(45.2) | 4(5.5) | 9(12.3) | 21(28.8) | 5(6.8) | 0(0) | 1(1.4) | 0(0) | 73(100) |
| Northeast | 0(0) | 4(36.4) | 1(9.1) | 1(9.1) | 4(36.4) | 1(9.1) | 0(0) | 0(0) | 0(0) | 11(100) |
| East | 3(0.8) | 187(50.3) | 30(8.1) | 49(13.2) | 46(12.4) | 37(9.9) | 13(3.5) | 1(0.3) | 6(1.6) | 372(100) |
| Central South | 15(4.1) | 105(28.9) | 65(17.9) | 14(3.9) | 53(14.6) | 68(18.7) | 10(2.8) | 3(0.8) | 30(8.3) | 363(100) |
| Southwest | 44(6.2) | 70(9.9) | 6(0.8) | 123(17.4) | 281(39.7) | 20(2.8) | 95(13.4) | 63(8.9) | 5(0.7) | 707(100) |
| Northwest | 2(0.9) | 98(42.8) | 1(0.4) | 74(32.3) | 51(22.3) | 2(0.9) | 1(0.4) | 0(0) | 0(0) | 229(100) |
| Co-infection[e] | | | | | | | | | | |
| HCV single infection | 0(0) | 25(34.2) | 13(17.8) | 0(0) | 0(0) | 13(17.8) | 0(0) | 0(0) | 22(30.1) | 73(100) |
| HCV/HIV co-infection | 34(4.2) | 219(26.9) | 22(2.7) | 177(21.7) | 227(27.9) | 51(6.3) | 43(5.3) | 37(4.5) | 5(0.6) | 815(100) |

Data are n (%)

[a]Data for n = 638

[b]Data for n = 493

[c]Data for n = 1378

[d]Data for n = 1755

[e]Data for n = 886

North = Beijing, Hebei, Shanxi, Inner Mongolia

Northeast = Liaoning, Heilongjiang

East = Shanghai, Jiangsu, Zhejiang, Anhui, Jiangxi, Shandong

Central South = Henan, Hubei, Hunan, Guangdong,Guangxi, Hainan

Southwest = Chongqing, Sichuan, Guizhou, Yunnan

Northwest = Shannxi, Qinghai, Sinkiang

MSM = Men who have sex with men

PWID = People who inject drugs

Other = 6e, 6 g, 6 l, 6w, and 6v.

(clusters) comprising 530 nodes (individual sequence) and 2,194 edges (undirected, potential links). The average degree (number of edges per node) was 4.1. The number of sequences per cluster ranged from 2–84 (median: 3, interquartile range:2–3) (Fig 2). In multivariable logistic analyses, being in a cluster was significantly associated with region (OR: 0.37, 95% CI: 0.19–

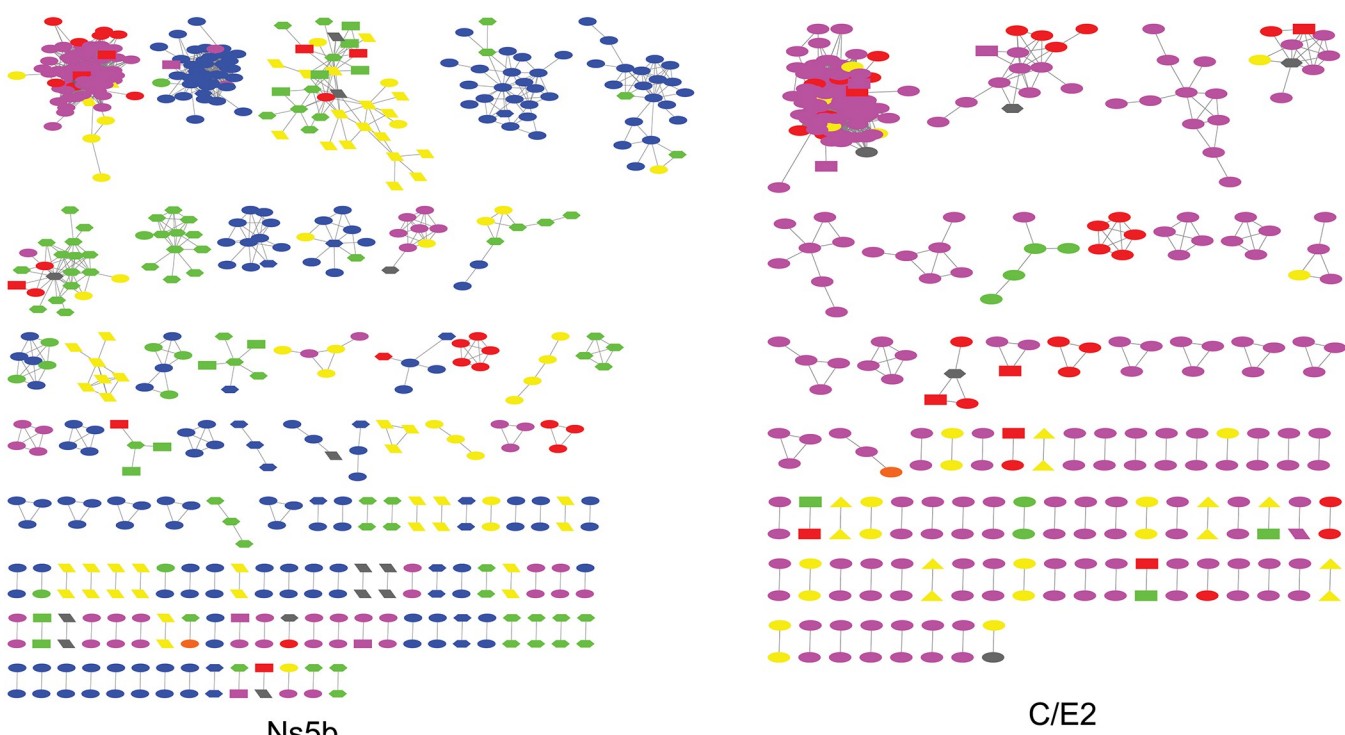

Ns5b

C/E2

**Fig 2. HCV molecular transmission network in China.** Clusters with ≥2 cases (i.e., nodes) are depicted. Links (i.e., edges) indicate genetic distance≤0.01 substitutions/site for *Ns5b* and ≤0.0325 substitutions/site for C/E2. Shape indicates population groups: Diamond, heterosexual; ellipse, people inject drugs; rectangle, former blood donors; hexagon, unknown; triangle, general population; parallelogram, volunteer blood donors. Color indicate sampling region: Red, North; orange, Northeast; yellow, East; green, Central South; blue, Southwest; purple, Northwest. North = Beijing, Hebei, Shanxi, Inner Mongolia, Northeast = Liaoning, Heilongjiang, East = Shanghai, Jiangsu, Zhejiang, Anhui, Jiangxi, Shandong, Central South = Henan, Hubei, Hunan, Guangdong, Guangxi, Hainan, Southwest = Chongqing, Sichuan, Guizhou, Yunnan, Northwest = Shannxi, Qinghai, Sinkiang.

0.71), subtype (OR: 0.23, 95% CI: 0.1–0.52), and sampling period(OR: 0.43, 95% CI: 0.27–0.68) (S2 Table).

We repeated the same network inference procedure for 865 *C/E2* sequences. Although the available dataset is relatively smaller, we observed a similar pattern in the transmission network inferred using *C/E2* sequences (Fig 2 and S3 Table).

## Phylodynamic analyses and inference of divergence date

We performed a Bayesian SkyGrid Plots (BSP) analysis for 19 datasets: 1) six *Ns5b* datasets (1a, 1b, 2a, 3a, 3b, 6a, 6n, and 6xa), 2) six *C/E2* datasets (1a, 1b, 2a, 3a, 3b, 6a, 6n, and 6xa), and 3) three *Ns5b* +*C/E2* concatenated datasets (1b, 3a, and 3b). Table 3 and S3 Fig. show the date of the Time to the Most Recent Common Ancestor (TMRCA) for the eight major HCV subtypes. Among them, subtype 1a and 6n were the oldest, subtype 6xa was the youngest. The TMRCA dates for strains 1b, 2a, 3a, and 3b were in the same range, approximately 80 years ago. The BSP shown in Figs 3 and S4. depict the estimated change in the effective number of infected individuals over time. Of the eight major subtypes, the epidemic history of 1b was one of most complicated in our datasets: it showed an "M-shape" or "Roller Coaster" curve that consisted of two major epidemic waves. The first wave began circa 1910 and ended circa 1985, with a peak circa 1957. The increasing period of the wave coincides with the introduction of modern medicine in China (probably through the reuse and inadequate sterilization of glass and metal syringes). The decreasing period coincides with the two major social and political events in

**Table 3. The TMRCA of HCV in China.**

|  | *Ns5b* |  | *C/E2* |  | *C/E2+ Ns5b* |  |
|---|---|---|---|---|---|---|
| Subtype | Numbers |  | Numbers |  | Numbers |  |
| 1a | 58 | 1899.6(1616.1–1981.5) | 16 | 2007.4(1960–2013.1) |  |  |
| 1b | 458 | 1915.2(1871.8–1942) | 282 | 1920.4(1897.4–1939.3) | 128 | 1897.3(1844.9–1935.4) |
| 2a | 95 | 1977.9(1963–1987.8) | 33 | 1800.2(190–1975.9) |  |  |
| 3a | 216 | 1948.9(1910.3–1974) | 143 | 1959.9(1947.9–1970.3) | 75 | 1955.4(1935.9–1971) |
| 3b | 399 | 1952.2(1918.8–1976.2) | 188 | 1962.9(1949.9–1974.7) | 69 | 1975.9(1961.7–1985.8) |
| 6a | 112 | 1964.6(1938.2–1982.2) | 44 | 1972.3(1954.1–1984.5) |  |  |
| 6n | 80 | 1967.8(1925–1993.5) | 50 | 1932.3(1836.5–1973.7) |  |  |
| 6xa | 58 | 1972.9(1935.5–1990.5) | 42 | 1976(1957.6–1988.5) |  |  |

TMRCA = Time to the Most Recent Common Ancestor.

aData are TMRCA (the 95% highest posterior density [HPD] interval).

China: the "Great Leap Forward" (1958–1960) and "Cultural Revolution" (1966–1976). The second wave seemed to be sparked by the increase in PWID in the middle 1980s and was enhanced by the "Encouraged Plasma Campaign"(1993–2000) in the 1990s. This escalating trend was abruptly reversed in approximately 2000, when the Chinese government outlawed

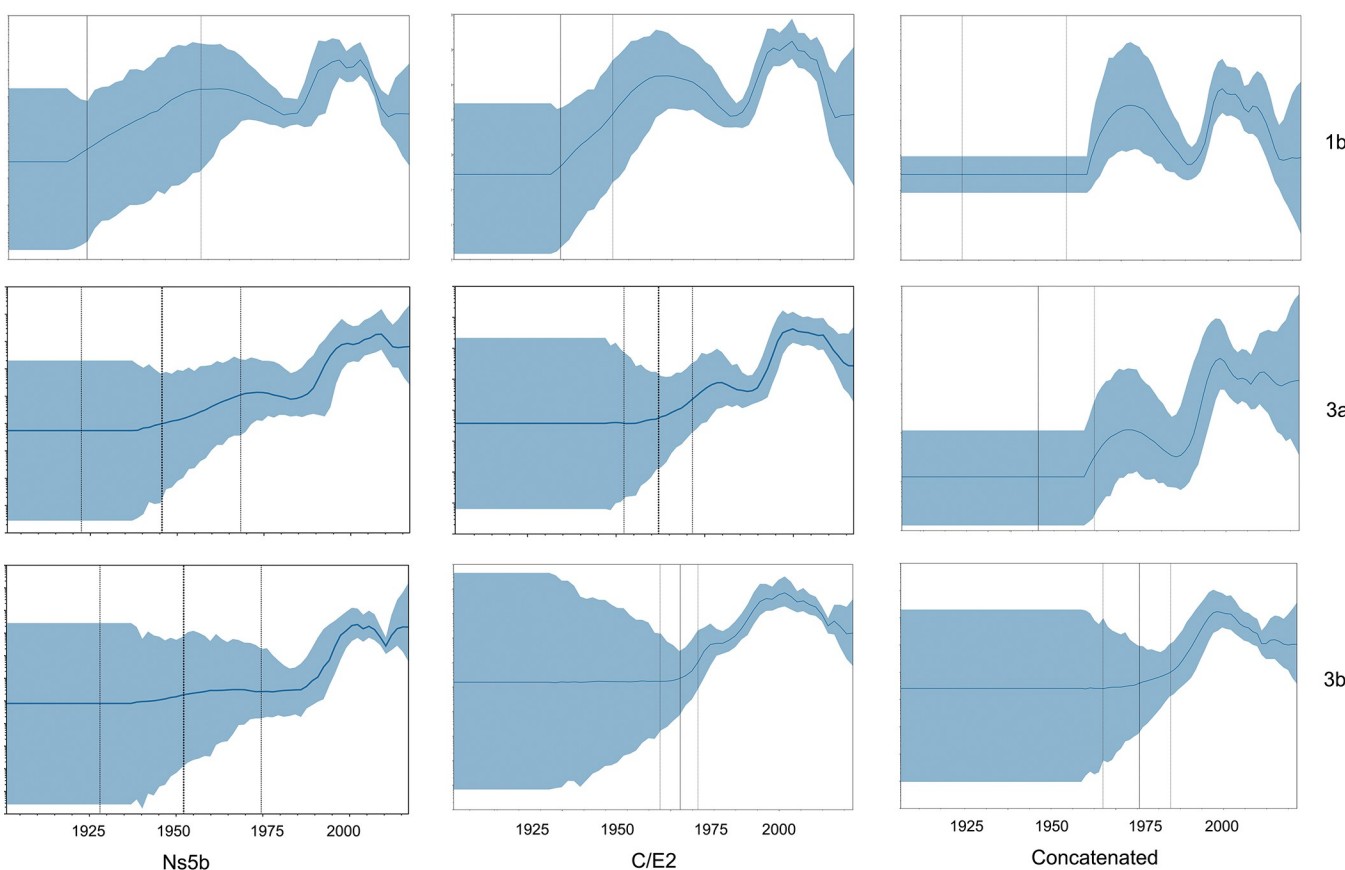

**Fig 3. The past population dynamics of HCV visualized using the Skygrid model.** The shaded portion is the 95% Bayesian credibility interval, and the solid line is the posterior median.

the use of paid blood donors. After that, despite a small rebound between 2005 and 2010, the 1b epidemic entered a downward trend from 2010 until the present. The other seven major subtypes have similar but relatively simple BSP curves. Of note, five major subtypes (1a, 1b, 2a, 3a, and 3b) exhibited a declining trend after 2010 until the present, whereas three subtypes (6a, 6n and 6xa) showed an increasing or stable trend. We repeated the same phylodynamic procedure using *C/E2* sequences datasets, and TMRCA and BSP were roughly consistent with that of *Ns5b* except for subtype 1a (Figs 3 and S4). The BSP for concatenated datasets have smaller confidence limits but narrower time scale (Fig 3).

## Discussion

Here, we report large amounts of data on HCV molecular epidemiology in China based on demographic and clinical data and HCV sequences from 1,811 patients of 25 provinces between 1994 and 2020. These data show that the HCV epidemic in China exhibits some degree of genetic diversity [34–41], consisting of four HCV subtypes and corresponding to 13 subtypes. Consistent with previous studies, the most prevalent HCV variant was subtype 1b, followed by 3b and 3a [6–11]. These subtypes are responsible for the majority of HCV cases globally [34–41]. Of note, five of eight major epidemic subtypes, together with 81.8% of HCV strains in our study, showed a declining tendency in effective population size during the past decade. In HCV transmission network analysis, 33.1% of patients grouped into 111 molecularly defined HCV transmission clusters.

Nakano, et al. [12] also reconstructed the population genetic history of HCV 1b in China and found that both groups of 1b grew at a rapid exponential rate during the "Cultural Revolution" of 1966–1976. They further attribute this rapid growth to the introduction of a million nonprofessional health-care providers ("barefoot doctors"). Barefoot doctors were healthcare providers who underwent basic medical training and worked in rural villages in China. The barefoot doctors system was developed and institutionalized in 1965 and broke down in the 1980s. Barefoot doctors included farmers, folk healers, rural healthcare providers and recent middle or secondary school graduates who received minimal basic medical and paramedical education.

Contrary to Nakano's finding, we observed a declining trend for HCV 1b in the effective population size during the "Cultural Revolution", and we suggest that attributing the increasing trend only to the introduction of "barefoot doctors" during the "Cultural Revolution" is oversimplified. Indeed, the impacts of large historical events such as the "Cultural Revolution" on the epidemic dynamics of HCV are complicated. On the one hand, the closure of medical schools and specialist hospital departments led to the introduction of "barefoot doctors" into the medical system, which may have caused an increase in HCV infection. On the other hand, nearly all professional medical staff had to stop working and were dispersed across the countryside during that period, which led to a sharp decline in the total amount of medical activity, including unsafe injections. We believe that the latter was the real determinant for the declining trend in the "Cultural Revolution" period.

Pybus et al. [42] showed that genotype 6 infection worldwide descended from a common ancestor that existed approximately 1,100 to 1,350 years earlier. How stable endemic transmission of HCV genotype 6 could be maintained for such long a time period has always fascinated scientists. As introduction and transmission events of HCV genotype 6 occurred so many years ago, we can only speculate. We suggest that traditional tattooing, which once prevailed in some minor ethnic populations of Yunnan Province, is responsible [43]. We further suggest that Yunnan is the epidemic centre of HCV genotype 6 in China as well as that of HIV [44–46]. Yunnan is located in southwestern China, bordering Myanmar, Laos, and Vietnam. There

are 16 ethnic minorities inhabiting the border, many of whom used to practice the custom of a traditional tattooing. The proximity and close cultural ties between populations in Yunnan and Southeast Asia countries have linked these groups for many years. It is plausible to speculate that HCV genotype 6 was introduced to China from Southeast Asian and maintained through traditional tattooing until the modern time, when this traditional custom was no longer popular.

To our knowledge, this is the largest study of its type thus far and involves the longest time period. Through this informative dataset, we conducted a national HCV molecular epidemiology study with broad representativeness and accurate phylogenetic reconstruction.

This analysis also has limitations. First, since approximately two-thirds of the sequences were from publicly available databases, most of the baseline characteristics of the patients were not available, which prevented us from including these variables in transmission cluster analysis and from making a more detailed investigation of the risk factors driving HCV epidemics in China. Second, because we used a cost-effective sampling method, participants with HIV/ HCV co-infection or PWID were overrepresented in our study. Hence, the findings might not be fully representative of the whole population in China. Third, the number of the recently sampled sequences was relatively small (S5 Fig). Therefore, the small rebound observed between 2005 and 2010 in our study is more likely due to sampling biases (e.g., distribution of samples in time and, lack of convergence in chain) than a real trend in the data.

Fourth, we discarded 152(6.5%) sequences from the original dataset because we thought they had quality problems, which could reshape a dataset with no temporal signal into one that strongly supports phylogenetic molecular clock analysis. The original results without the filtering sequences are listed in S4 Table.

In summary, this national study of 1,811 patient HCV sequences describes the most recent data on HCV genotype distribution in China. The most common HCV strain was found to be 1b, followed by 3b and 3a. Phylodynamic analysis revealed a complex scenario that was most likely driven by a combination of social, demographic, and medical factors over both recent and historical timescales. Crucially, BSP analysis showed a declining trend up to the present for 81.8% of the HCV strains in our study, which is a good omen for the goal of eliminating HCV by 2030.

## Supporting information

**S1 Fig. Number of transmission clusters as a function of the TN93 distance.** The threshold that was selected is highlighted in red.
(DOCX)

**S2 Fig. The maximum likelihood (ML) phylogenetic tree based on Ns5b and C/E2 gene.** 1,603 Ns5b and 865 C/E2 sequences from China were analyzed with HCV reference strains (NC_004102, D90208, AB047639, JN714194, JQ065709, HQ639936, DQ278894) as an outgroup using Fasttree 2.1.
(DOCX)

**S3 Fig. The TMRCA of HCV in China.** TMRCA = Time to the Most Recent Common Ancestor. The solid line indicates the 95% highest posterior density [HPD] interval for TMRCA.
(DOCX)

**S4 Fig. The past population dynamics of HCV (1a, 2a, 6a, 6n, and 6xa) visualized using the Skygrid model.** The shaded portion is the 95% Bayesian credibility interval, and the solid line is the posterior median.
(DOCX)

**S5 Fig. The distribution of sampling year for HCV sequences in China.**
(DOCX)

**S1 Table. Descriptions of the participating cohort.TDR = Transmitted drug resistance.**
BHLN = Beijing HIV laboratory network; PWID = People who inject drugs; MSM = Men who
have sex with men; NA = not available; LANL = Los Alamos National Laboratory.
(DOCX)

**S2 Table. Demographic and clinical factors associated with clustering based on Ns5b gene.**
North = Beijing, Hebei, Shanxi, Inner Mongolia, Northeast = Liaoning, Heilongjiang,
East = Shanghai, Jiangsu, Zhejiang, Anhui, Jiangxi, Shandong, Central South = Henan, Hubei,
Hunan, Guangdong,Guangxi, Hainan, Southwest = Chongqing, Sichuan, Guizhou, Yunnan,
Northwest = Shannxi, Qinghai, Sinkiang; MSM = men who have sex with men,
PWID = people who inject drugs; NA = not available; OR = odds ratio; aData are n (%); bUni-
variable logistic regression analysis; cMultivariable logistic regression analysis; dData for
n = 1552, eData for n = 1175, Other = 6e, 6g, 6l, 6w, and 6v.
(DOCX)

**S3 Table. Demographic and clinical factors associated with clustering based on C/E2 gene.**
North = Beijing, Hebei, Shanxi, Inner Mongolia, Northeast = Liaoning, Heilongjiang,
East = Shanghai, Jiangsu, Zhejiang, Anhui, Jiangxi, Shandong, Central South = Henan, Hubei,
Hunan, Guangdong,Guangxi, Hainan, Southwest = Chongqing, Sichuan, Guizhou, Yunnan,
Northwest = Shannxi, Qinghai, Sinkiang; MSM = men who have sex with men,
PWID = people who inject drugs; NA = not available; OR = odds ratio; aData are n (%), bUni-
variable logistic regression analysis, cMultivariable logistic regression analysis, dData for
n = 849, eData for n = 846, Other = 6e, 6g, 6l, 6w, and 6v.
(DOCX)

**S4 Table. The TMRCA of HCV in China inferred from original dataset.** TMRCA = Time to
the Most Recent Common Ancestor. aData are TMRCA (the 95% highest posterior density
[HPD] interval).
(DOCX)

**S1 File. The protocol for Bayesian estimation of past population dynamics using the Sky-
grid coalescent model.**
(DOCX)

**S2 File. Accession numbers.**
(DOCX)

## Acknowledgments

We thank the study participants and the staff at the collaborating clinical sites and laboratories.
We thank the local health workers of the BHLN, who spent numerous hours and great effort
in obtaining, verifying, and cleaning the data used in this study. We thank Dr. Xiang He from
Guangdong Institute of Public Health for useful comments on drafts of the manuscript.

## Author Contributions

**Conceptualization:** Yiming Shao, Hui Xing, Hongyan Lu.

**Data curation:** Jingrong Ye, Jia Li.

**Formal analysis:** Jingrong Ye, Xinli Lu, Minna Zheng.

**Funding acquisition:** Jingrong Ye.

**Investigation:** Yanming Sun, Shufang He, Conghui Xu, Xianlong Ren, Juan Wang, Jing Chen.

**Methodology:** Jingrong Ye, Jia Li.

**Resources:** Lifeng Liu, Fengting Yu.

**Software:** Jingrong Ye.

**Supervision:** Hongyan Lu.

**Validation:** Yuhua Ruan, Yi Feng.

**Writing – original draft:** Jingrong Ye.

**Writing – review & editing:** Jingrong Ye, Hui Xing, Hongyan Lu.

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
