## [Decision Letter · Decision Letter 0]

27 Feb 2023

PONE-D-22-30314Distribution pattern, phylodynamic and molecular transmission networks of hepatitis C virus in ChinaPLOS ONE

Dear Dr. Lu,

Thank you for submitting your manuscript to PLOS ONE. After careful consideration, we feel that it has merit but does not fully meet PLOS ONE’s publication criteria as it currently stands. Therefore, we invite you to submit a revised version of the manuscript that addresses the points raised during the review process.

We look forward to receiving your revised manuscript.

Kind regards,

Jason T. Blackard, PhD

Academic Editor

PLOS ONE

Journal Requirements:

3. We note that Figure (2) in your submission contain copyrighted images. All PLOS content is published under the Creative Commons Attribution License (CC BY 4.0), which means that the manuscript, images, and Supporting Information files will be freely available online, and any third party is permitted to access, download, copy, distribute, and use these materials in any way, even commercially, with proper attribution. For more information, see our copyright guidelines: http://journals.plos.org/plosone/s/licenses-and-copyright.

1. You may seek permission from the original copyright holder of Figure (2) to publish the content specifically under the CC BY 4.0 license. 

Additional Editor Comments (if provided):

This a large cross-sectional study of HCV transmission networks in 4 surveillance populations in China.

Overall, the methods and results are well described.

In addition to addressing the comments raised by the two reviewers, the revisions below would strengthen the manuscript further.

The authors mention duplicate sequences and repeated sampling of the same individuals but never state how often these two situations actually occurred here.Table 3 can be simplified by rounding to the nearest whole year.

Reviewers' comments:

Reviewer's Responses to Questions

**Comments to the Author**

1. Is the manuscript technically sound, and do the data support the conclusions?

Reviewer #1: Partly

Reviewer #2: Yes

2. Has the statistical analysis been performed appropriately and rigorously? 

Reviewer #1: Yes

Reviewer #2: Yes

3. Have the authors made all data underlying the findings in their manuscript fully available?

Reviewer #1: No

Reviewer #2: Yes

4. Is the manuscript presented in an intelligible fashion and written in standard English?

Reviewer #1: Yes

Reviewer #2: Yes

5. Review Comments to the Author

Reviewer #1: This paper reports a fairly conventional comparative study of HCV sequences that were collected in China. Like many previous studies, it uses BEAST to reconstruct changes in coalescence rates (i.e., effective number of infections) over time with time-stamped sequences. They also use pairwise genetic distances to cluster sequences into connected components and then evaluate statistical associations of this outcome with individual-level variables such as transmission risk. The authors distinguish their study from previous work by the number of sequences, and the addition of clinical sequences with metadata such as mode of transmission. Overall the manuscript is quite well written.

First, I have some concerns about how the BEAST analyses were carried out. It is generally considered good practice to assess whether there is adequate evidence of a molecular clock in your data by root-to-tip regression (e.g., TempEST) before attempting an analysis in BEAST. This may impact some of the HCV subtypes with smaller numbers of sequences.

The results would also be more convincing if the authors had run replicate chains for each dataset to demonstrate convergence. Furthermore, wouldn't it be possible to analyze both NS5B and C/E2 regions jointly? There is limited recombination in HCV, so combining the sequences should lend more statistical power to estimate TMRCA etc. The sequences can be concatenated, adding stretches of N's where one region is not covered in a given sample. On the other hand, adding the LANL data severely reduces the proportion of samples for which both regions are covered.

The authors should include a figure summarizing the distribution of sample collection dates over time as a histogram or density plot - this could even be incorporated into the Bayesian skyline plots. In addition, the skyline plots should be moved into the main document, ideally displayed with the same time (horizontal) axis. Moreover, the authors should consider combining skylines into a single plot.

Please report the configuration of BEAST analyses (e.g., prior hyperparameters) to a degree that would enable others to reproduce the analysis. Please report specific criteria used to assess convergence, such as effective sample size.

Specific comments:

- page 3, line 56 abstract and elsewhere, "65.2% of HCV strains" - it is not clear what the authors mean by strains. Are they referring to the different HCV subtypes / genotypes analyzed independently in BEAST?

- page 4, lines 87-88: "However, the outcomes of these theoretical studies have been limited by a relatively narrow span of sampling time." What makes these past studies theoretical? Nakano et al. (2006) and Lu et al. (2013) report phylogenetic analyses of HCV sequence data, so they seem just as empirical as the present study.

- page 5, line 94, please clarify what you mean by "epidemiologic connection"

- page 5, lines 95-96: "Over the last decade, a simplified genetic distance-based method has increasingly been used to define HIV transmission networks within a population." The authors should indicate which method they are talking about (there are several), and cite the relevant literature.

- page 5, line 105: "[...] using our unique dataset." Every dataset is unique. It would be more appropriate to state "[...] using a substantially more comprehensive dataset and metadata than previous work." or something along those lines.

- page 6, line 111: please define BHLN, CDC and PLA at first use.

- page 6, line 114: do the authors mean "reference laboratory" when they write "confirmatory laboratory"? I have not seen this term used as a noun in the literature - I can only find the phrase "confirmatory laboratory testing".

- page 6, line 119 and elsewhere: perhaps use "cost-effective" rather than "economical"?

- page 6, lines 123-124: please cite a reference

- page 6, line 130: "Accepting the reality" - this is awkward phrasing

- page 6, line 131: "we devised unique economic [...]" - is it really necessary to assert that these inclusion criteria are unique?

- page 7, it would be helpful to refer to Figure 1 (data collection flowchart) somewhere here, and/or Table S1

- lines 169-170: presumably building the ML tree was to confirm COMET geno/subtyping - please clarify

- lines 173-174: how many duplicate sequences? It is not necessary to discard these for a molecular clock analysis (i.e., BEAST), and in fact removing duplicate sequences can bias the analysis.

- line 177: what do you mean by "miscued"? Do you mean "misclassified"?

- lines 188-189: the authors report generating maximum clade credibility trees, but these do not appear anywhere in the manuscript or supplementary materials.

- lines 195: what is the rationale for using these TN93 distance thresholds? How sensitive are your results to varying these thresholds?

- lines 219-222: minor point - it is a bit unusual to report both confidence intervals and P-values, where CI is the presently the recommended method for assessing statistical significance.

- lines 261-262: please display the phylogenetic tree as a supplementary figure, preferably with tips coloured by COMET results

- line 272: "Table 2 presents the temporal trends for these eight major subtypes." Table 2 does not appear to contain any temporal information - I think the authors meant to refer to Table 1?

- Please consider using a set of choropleths (e.g., https://www.esri.com/arcgis-blog/wp-content/uploads/2020/02/redmap.png) to summarize the distribution of HCV subtypes in China (one choropleth per subtype) instead of pie charts. I think they would be much easier to interpret.

- Table 3 might be more effectively presented as a series of histograms or density plots (e.g., ridgeplots)

- line 287: regarding "modern medicine", it is most likely the reuse and inadequate sterilization of glass and metal syringes, is it not?

- lines 293-294: the "small rebound between 2005 and 2010" is more likely due to sampling biases (e.g., distribution of samples in time, lack of convergence in chain) than a real trend in the data.

- lines 298-300: it is not sufficient to claim that the BEAST analyses of C/E2 sequences "were consistent with that of NS5B" without showing any data or reporting any quantitative results. It would not be difficult, for example, to plot the skylines for both genome regions together for a given subtype.

- lines 310-312: Please summarize odds ratios and CIs from univariate analyses here. The reader should not have to dig into the supplementary materials for this information.

- lines 314-316: "Although the available dataset is relatively smaller, we observed a similar pattern in the transmission network inferred using C/E2 sequences." This is really inadequate. If you are not going to show these results (e.g., sizes and composition of largest components, network graph), then you need to justify this conclusion with quantitative results, i.e., a statistical comparison of the two networks.

- lines 321-322: "These data show that the HCV epidemic in China exhibits great genetic diversity." This claim needs to be justified. How much more diverse are the HCV sequences (with respect to number of different genotypes and subtypes) in China compared to other regions? Ideally you should adjust for differences in sample sizes.

- line 331 and line 349: It is unconventional to give the full name of the first authors when referencing previous work. Usually one would just write "Nakano et al.", for example.

- lines 334-344: This section really needs supporting references to the peer-reviewed literature.

- line 347: typo, should be "Asia".

- line 575: "All code is shared openly for review." Where? Please provide a URL.

- lines 576-577: "HCV sequences have been submitted to Genbank." Are the accession numbers available? Please provide them.

- The legend for Figure 1 is very incomplete, as is Figure 2.

- lines 609-610: the highest posterior density (Bayesian) is not equivalent to a confidence interval (frequentist).

- Table 3, why are some tMRCA estimates not available ("NA")?

- Figure 3 axis label, "Cultural Revolution", not "Culture Revolution"

- Supplementary methods, text on Hukou system is not referenced in main text.

Reviewer #2: This is an interesting article characterizing HCV phylogenetic analysis in China. Overall, the manuscript would benefit from more background/detail as outlined below.

Line 247: The authors note the median baseline CD4 count was 336. Was this only among people living with HIV? This should be clarified.

Line 253: How did the authors arrive at 1024 and 1811? It doesn't seem like this number is possible given the samples available based on the text in the manuscript. It becomes evident when looking at figure 1. Better characterization of the number of samples obtained from the LANL database would be helpful.

Figure 1: How are the authors able to define the "effective sample size"? It would be helpful to define this, explicitly in the methods.

Line 334-338: The authors should unpack terms like barefoot doctors and the importance of the Cultural Revolution on HCV spread more for those who aren't familiar with this literature. A discussion of these factors and what is known about them thus far in the literature might fit nicely in the introduction.

Line 353: How did the authors arrive at 200 years? Again, better description in the method section of how these estimates are made would be beneficial.

Line 359: What is the link between HIV, HCV, and Yunnan. It seems plausible that HCV could be spread by traditional tattooing, but this seems unlikely for HIV. Is HIV thought to have originated in China in this area? Was this conclusion arrive at through phylogenetic analysis? Again, more thorough explanation would be useful.

Minor points:

- Please use person first language – e.g. people who inject drugs rather than IDU

6. PLOS authors have the option to publish the peer review history of their article (what does this mean?). If published, this will include your full peer review and any attached files.

Reviewer #1: **Yes: **Art Poon

Reviewer #2: No

---

## [Author Response · Author response to Decision Letter 0]

26 Jun 2023

Dear Editor and Reviewers:

We are very grateful to you for giving us an opportunity to revise our manuscript. We really appreciate you very much for your positive and constructive comments and suggestions on our manuscript. We have studied reviewers' comments carefully and tried our best to revise our manuscript according to the comments.

The following are the responses and revisions we have made in response to the reviewer' questions and suggestions point-by-point.

Thanks again to the hard work of the editor and reviewers.

Reviewer #1: This paper reports a fairly conventional comparative study of HCV sequences that were collected in China. Like many previous studies, it uses BEAST to reconstruct changes in coalescence rates (i.e., effective number of infections) over time with time-stamped sequences. They also use pairwise genetic distances to cluster sequences into connected components and then evaluate statistical associations of this outcome with individual-level variables such as transmission risk. The authors distinguish their study from previous work by the number of sequences, and the addition of clinical sequences with metadata such as mode of transmission. Overall the manuscript is quite well written.

Dear Professor Poon:

Response: Thank you for the positive comments for our manuscript.

First, I have some concerns about how the BEAST analyses were carried out. It is generally considered good practice to assess whether there is adequate evidence of a molecular clock in your data by root-to-tip regression (e.g., TempEST) before attempting an analysis in BEAST. This may impact some of the HCV subtypes with smaller numbers of sequences.

Response: Thank you for your good suggestion. We assessed all the sequences using TempEst and found that there were 146 sequences with quality problems. After excluding the 146 sequences, we went back and reanalyzed all the data and got completely different result. We should have known TempEst earlier.

The results would also be more convincing if the authors had run replicate chains for each dataset to demonstrate convergence. Furthermore, wouldn't it be possible to analyze both NS5B and C/E2 regions jointly? There is limited recombination in HCV, so combining the sequences should lend more statistical power to estimate TMRCA etc. The sequences can be concatenated, adding stretches of N's where one region is not covered in a given sample. On the other hand, adding the LANL data severely reduces the proportion of samples for which both regions are covered.

Response: We ran replicated the chains for each datasets and combined the result using the LogCombiner. The performance of ESS value improved a lot. The Skygrid results for concatenated datasets have smaller confidence limits but narrower time scale. As expected, few patients from LANL have sequences covering both Ns5b and C/E2 region. Furthermore, even both Ns5b and C/E2 were available for one patient, we were not sure that they are from identical patient. Therefore, we constructed concatenated dataset for Ns5b and C/E2 sequences only from our study. We succeeded in getting 3 concatenated datasets for 1b, 3a, and 3b, but not for other subtype (1a, 2a, 6a, 6n, and 6xa), because the number of the sequences is too small. 

The authors should include a figure summarizing the distribution of sample collection dates over time as a histogram or density plot - this could even be incorporated into the Bayesian skyline plots. In addition, the skyline plots should be moved into the main document, ideally displayed with the same time (horizontal) axis. Moreover, the authors should consider combining skylines into a single plot.

Response: As you suggested, we included a histogram which summarize the distribution of sampling year. However, we did not incorporated the histogram in the Bayesian Skygrid plot because the Skygrid plots were too busy. We provided the histogram in the supplementary materials. The Skygrid plot was moved into the main document, which was displayed with the same time axis.

Please report the configuration of BEAST analyses (e.g., prior hyperparameters) to a degree that would enable others to reproduce the analysis. Please report specific criteria used to assess convergence, such as effective sample size.

Response: We provided a SOP for the BEAST analysis in supplementary method section, which enable anyone to reproduce our analysis. We also provided the criteria for assessing the convergence in the method section.

Specific comments:

- page 3, line 56 abstract and elsewhere, "65.2% of HCV strains" - it is not clear what the authors mean by strains. Are they referring to the different HCV subtypes / genotypes analyzed independently in BEAST?

Response: 65.2% was updated as 81.8%. In this study, we reconstruct the past dynamic history for 8 HCV subtype. The results of Skygrid showed that five of them (1a, 1b, 2a, 3a, and 3b), which constituted 81.8% (1447 of 1769) of HCV strains genotyped, have declining trend. 

- page 4, lines 87-88: "However, the outcomes of these theoretical studies have been limited by a relatively narrow span of sampling time." What makes these past studies theoretical? Nakano et al. (2006) and Lu et al. (2013) report phylogenetic analyses of HCV sequence data, so they seem just as empirical as the present study.

Response: The studies by professor Nakano,et al. and professor Lu et al. were just classic. I learn a lot from these two articles. However, even the most recent study was conducted ten years ago. The technology of phylodynamic analysis are progressing. The data about molecular epidemiology of HCV in China need updated. We provided a table (Table 1) to summary the characteristic of these two studies and ours.

Table 1. The characteristic of studies by Nakano,et al., Lu et al., and ours.

 Nakaro’ Lu’ Our

Gene E1 and Ns5b E1 and Ns5b Ns5b and C/E2

Sampling region 9 cities# 1 city (Guangzhou) 25 provinces*

Sampling period 2002 2009-2011 1994-2020

Number of sequence 

E1 89 417 

Ns5b 92 423 406

C/E2 397

Number of reference sequence 

E1 72 

Ns5b 61 1197

C/E2 468

Subtype 1b 1b, 2a, 3a, 3b, 6a 1a, 1b, 2a, 3a, 3b, 6a, 6n, and 6xa

Model Bayesian skyline plots in BEAST 1.2. Bayesian skyline plots in BEAST 1.6.1. Bayesian Skygrid in BEAST 1.10.4.

#Shenyang, Beijing, Hohhot, Shanghai, Zhengzhou, Guangzhou, Shenzhen, Foshan, Kunming.

*North=Beijing, Hebei, Shanxi, Inner Mongolia; Northeast=Liaoning, Heilongjiang; East=Shanghai, Jiangsu, Zhejiang, Anhui, Jiangxi, Shandong; Central South=Henan, Hubei, Hunan, Guangdong,Guangxi, Hainan; Southwest=Chongqing, Sichuan, Guizhou, Yunnan; Northwest=Shannxi, Qinghai, Sinkiang.

- page 5, line 94, please clarify what you mean by "epidemiologic connection"

Response: In this article, “epidemiologic connection” means “epidemiologically related”. For example, two individuals with similar viruses are likely to have direct transmission relationship, or be infected by a common source. A cluster of individuals with genetically similar infections may represent a outbreak related through a succession of recent transmission events.

- page 5, lines 95-96: "Over the last decade, a simplified genetic distance-based method has increasingly been used to define HIV transmission networks within a population." The authors should indicate which method they are talking about (there are several), and cite the relevant literature.

Response: To date, as far as I knew, there is still no clear consensus on how transmission clusters should be defined. Over the last two decades, many clustering methods have been developed to define HIV transmission networks within a population. Broadly speaking, these methods can be grouped into two categories: methods that cluster directly on sequence variation via pairwise genetic distance measures, and methods that interpret this variation in the context of subtrees in a phylogeny. Phylogenetic analysis can be associated with high computational burden, especially for large sequence datasets. However, the genetic distance method can be computed rapidly. Therefore recent network analyses have favoured the generally faster and parameter-rich distanced-based methods. We chose to used the pairwise genetic method too.

- page 5, line 105: "[...] using our unique dataset." Every dataset is unique. It would be more appropriate to state "[...] using a substantially more comprehensive dataset and metadata than previous work." or something along those lines.

Response: I agreed with your opinion. We removed the word “unique” in our manuscript. 

- page 6, line 111: please define BHLN, CDC and PLA at first use.

Response: We gave the full name for these three abbreviation when they first appeared in the manuscript.

- page 6, line 114: do the authors mean "reference laboratory" when they write "confirmatory laboratory"? I have not seen this term used as a noun in the literature - I can only find the phrase "confirmatory laboratory testing".

Response: Yes, confirmatory laboratories are reference laboratories.

- page 6, line 119 and elsewhere: perhaps use "cost-effective" rather than "economical"?

Response: We used cost-effective in new revision.

- page 6, lines 123-124: please cite a reference

Response: We cited the reference.

- page 6, line 130: "Accepting the reality" - this is awkward phrasing

Response: We removed this phrase.

- page 6, line 131: "we devised unique economic [...]" - is it really necessary to assert that these inclusion criteria are unique?

Response: We removed the word “unique” in the new revision.

- page 7, it would be helpful to refer to Figure 1 (data collection flowchart) somewhere here, and/or Table S1

Response: We referred to Fig 1 and Table S1 .

- lines 169-170: presumably building the ML tree was to confirm COMET geno/subtyping - please clarify

Response: Yes, we built the ML tree to confirm the result from COMET.

- lines 173-174: how many duplicate sequences? It is not necessary to discard these for a molecular clock analysis (i.e., BEAST), and in fact removing duplicate sequences can bias the analysis.

Response: As the datasets were not too large, we realized that there is no need to discard duplicate sequences. In the new revision, we restored the sequences that were discarded, as you suggested.

- line 177: what do you mean by "miscued"? Do you mean "misclassified"?

Response: Yes, the “miscued” do mean the “misclassified”. We corrected it.

- lines 188-189: the authors report generating maximum clade credibility trees, but these do not appear anywhere in the manuscript or supplementary materials.

Response: In the new revision, we provided the maximum clade credibility(MCC) trees in supplementary materials (Fig S2).

- lines 195: what is the rationale for using these TN93 distance thresholds? How sensitive are your results to varying these thresholds?

Response: First, the TN93 genetic distance was used because it can be computed rapidly and is the most complex genetic distances that can be represented by a closed-form solution; Second, it is easy to grasp; Third, it is very popular. As far as I know, more than 80% article about HIV and HCV molecular transmission network published during the past decade used TN93 model. The sensitive analysis showed that the conclusion of the transmission network using different TN93 threshold did not changed.

- lines 219-222: minor point - it is a bit unusual to report both confidence intervals and P-values, where CI is the presently the recommended method for assessing statistical significance.

Response: We removed the P-values in the new revision.

- lines 261-262: please display the phylogenetic tree as a supplementary figure, preferably with tips coloured by COMET results.

Response: We provided the phylogentic trees as a supplementary figure (Figure S2).

- line 272: "Table 2 presents the temporal trends for these eight major subtypes." Table 2 does not appear to contain any temporal information - I think the authors meant to refer to Table 1?

Response: Yes, it referred Table 1. I corrected it.

- Please consider using a set of choropleths (e.g., https://www.esri.com/arcgis-blog/wp-content/uploads/2020/02/redmap.png) to summarize the distribution of HCV subtypes in China (one choropleth per subtype) instead of pie charts. I think they would be much easier to interpret.

Response: Including figure 2 in our manuscript brought some trouble to me.

The editor thought that Figure 2 may contain copyrighted images. The editor require me to either 1) present written permission from the copyright holder to publish these figures specifically under the CC BY 4.0 license, or 2) remove the figures from my submission. We preferred to remove it. We included this map to characterize the geographical distribution of HCV subtype in China. Under the present circumstance, we think it is better to characterize it in a table. 

I have heard so much about Arcgis for a long time. But I did not have opportunity to used it, because it is so expensive that I could not afford to buy it. 

- Table 3 might be more effectively presented as a series of histograms or density plots (e.g., ridgeplots)

Response: We transformed Table 3 as a series of histograms. But we still kept this table in the manuscript as well.

- line 287: regarding "modern medicine", it is most likely the reuse and inadequate sterilization of glass and metal syringes, is it not?

Response: Yes, I agreed with you for the opinion. We added it in the manuscript.

- lines 293-294: the "small rebound between 2005 and 2010" is more likely due to sampling biases (e.g., distribution of samples in time, lack of convergence in chain) than a real trend in the data.

Response: We agreed with you for the opinion. We included this opinion in the limitation section. 

- lines 298-300: it is not sufficient to claim that the BEAST analyses of C/E2 sequences "were consistent with that of NS5B" without showing any data or reporting any quantitative results. It would not be difficult, for example, to plot the skylines for both genome regions together for a given subtype.

Response: We have plot skygrid for both genome for the main eight subtype (1a, 1b, 2a, 3a, 3b, 6a, 6n, and 6xa). We also plot skygrid for concatenated genome for 1b, 3a, and 3b, but not for the other five subtype (1a, 2a, 6a, 6n, and 6xa) because the number of sequences are too small.

- lines 310-312: Please summarize odds ratios and CIs from univariate analyses here. The reader should not have to dig into the supplementary materials for this information.

Response: We summarized OR and CIs here.

- lines 314-316: "Although the available dataset is relatively smaller, we observed a similar pattern in the transmission network inferred using C/E2 sequences." This is really inadequate. If you are not going to show these results (e.g., sizes and composition of largest components, network graph), then you need to justify this conclusion with quantitative results, i.e., a statistical comparison of the two networks.

Response: We did the transmission network analysis in parallel using Ns5b and C/E2 and showed these results together in the new revision. 

- lines 321-322: "These data show that the HCV epidemic in China exhibits great genetic diversity." This claim needs to be justified. How much more diverse are the HCV sequences (with respect to number of different genotypes and subtypes) in China compared to other regions? Ideally you should adjust for differences in sample sizes.

Response: We adjusted this claim. As far as I knew, the number of sequences for most of study of this kind is about 500. Therefore we analyzed more than threefold of the numbers of sequences of other study.

- line 331 and line 349: It is unconventional to give the full name of the first authors when referencing previous work. Usually one would just write "Nakano et al.", for example.

Response: We made some adjustment.

- lines 334-344: This section really needs supporting references to the peer-reviewed literature.

Response: The peer-reviewed literature about relationship of HCV epidemiology and “Cultural Revolution” and “ Great Leap” were limited.

We searched HCV and/or “Cultural Revolution” or “Great Leap Forward” or “Encouraged Plasma Campaign” in Pubmed. We only got two articles. We just provided these two articles as supporting references.

- line 347: typo, should be "Asia".

Response: We corrected it.

- line 575: "All code is shared openly for review." Where? Please provide a URL.

Response: We provided a URL in the second revision (DOI 10.17605/OSF.IO/NKD8Y).

- lines 576-577: "HCV sequences have been submitted to Genbank." Are the accession numbers available? Please provide them.

Response:

We provided accession number of the sequences from LANL database in the supplementary materials. 

- The legend for Figure 1 is very incomplete, as is Figure 2.

Response: I apologized for my careless. In the new revision, we tried our best to make the legend as complete as possible.

- lines 609-610: the highest posterior density (Bayesian) is not equivalent to a confidence interval (frequentist).

Response: Thank you for your suggestion. We corrected it.

- Table 3, why are some tMRCA estimates not available ("NA")?

Response: We provided the missing tMRCA in the new revision.

- Figure 3 axis label, "Cultural Revolution", not "Culture Revolution"

Response: Thank you for your suggestion. We corrected it.

- Supplementary methods, text on Hukou system is not referenced in main text.

Response: Thank you for your suggestion. As Hukou system was not mentioned in discussion section, we have removed it in supplementary methods.

Reviewer #2: This is an interesting article characterizing HCV phylogenetic analysis in China. Overall, the manuscript would benefit from more background/detail as outlined below.

Dear Professor:

Response: Thank you for the positive comments for our manuscript.

Line 247: The authors note the median baseline CD4 count was 336. Was this only among people living with HIV? This should be clarified.

Response: 

Yes, the CD4 was only available for individuals with HIV/HCV co-infection.

We clarified it in the manuscript. This study is a secondary product of a multi-center HIV molecular epidemiology in China.

Line 253: How did the authors arrive at 1024 and 1811? It doesn't seem like this number is possible given the samples available based on the text in the manuscript. It becomes evident when looking at figure 1. Better characterization of the number of samples obtained from the LANL database would be helpful.

Response: I realized that there was much confusion in the numbers of sequences in the text. The flowchart also failed to give clear information indeed. We included 1,197 Ns5b, and 468 C/E2 sequences from 1343 individuals from the LANL database. Ns5b is over-represented. 322 individuals have both Ns5b and C/E2 sequences. 1,021 have either of the fragments, of which 875 have single Ns5b and 146 have single C/E2. We ourselves provided 406 Ns5b, and 397 C/E2 sequences from 468 individuals for this analysis. 335 individuals have both Ns5b and C/E2 sequences, and 133 have either of the fragments. Together we obtained 1603 Ns5b and 865 C/E2 sequences from 1811 individuals. In the pooled dataset, 657 have both regions and 1154 have either.

In the new revision, we excluded 3 C/E2 sequences with problem from LANL.

Therefore, number 1024 became 1021, but the 1811 was consistent.

There so many figures in the manuscript. We tried to present them clearly and concisely.

Figure 1: How are the authors able to define the "effective sample size"? It would be helpful to define this, explicitly in the methods.

Response: The Effective Sample Size (ESS) of a parameter sampled from an MCMC (such as BEAST) is the number of effectively independent draws from the posterior distribution that the Markov chain is equivalent to. We defined the ESS in methods section.

Line 334-338: The authors should unpack terms like barefoot doctors and the importance of the Cultural Revolution on HCV spread more for those who aren't familiar with this literature. A discussion of these factors and what is known about them thus far in the literature might fit nicely in the introduction.

Response:

Barefoot doctors were healthcare providers who underwent basic medical training and worked in rural villages in China. The barefoot doctors system was developed and institutionalized in 1965 and broke down in the 1980s. Barefoot doctors included farmers, folk healers, rural healthcare providers and recent middle or secondary school graduates who received minimal basic medical and paramedical education. The name comes from southern farmers in China, who would often work barefoot in the rice paddies, and simultaneously worked as medical practitioners. Major social and political events may deeply influence the transmission of infectious disease. “The culture revolution” is the largest social and political event in China during the past century. The cultural revolution damaged China’s healthcare system. During the revolution, nearly all professional medical staff had to stop working and were dispersed across the countryside.

Line 353: How did the authors arrive at 200 years? Again, better description in the method section of how these estimates are made would be beneficial.

Response: I am sorry for giving you the impression that we have time-travel to 200 years ago. Followed the suggestion given by professor Poon, we assess the quality of the sequences using the TempEst before doing analysis in BEAST. We excluded 146 problematic sequences and did BEAST analysis once more. At this time, most of TMRA fell within 100 years ago, except for subtype a1. 

Our study was inspired by two classic articles in HCV molecular epidemiology area. The first is the article entitled “Genetic history of hepatitis C virus in East Asia” wrote by professor Oliver published in J Virol (2009;83:1071-82.).

Oliver et al. revealed a >1,000-year-long development of genotype 6 in Asia, characterized by substantial phylogeographic structure and two distinct phases of epidemic history, before and during the 20th century. 

The second is article entitled “Colonial history and contemporary transmission shape the genetic diversity of hepatitis C virus genotype 2 in Amsterdam" wrote by professor Markov published in J Virol (2012;86:7677-87.).

Markov et al. detected multiple HCV-2 movements from present-day Ghana/Benin to the Caribbean during the peak years of the slave trade (1700 to 1850) and extensive transfer of HCV-2 among the Netherlands and its former colonies Indonesia and Surinam over the last 150 years. The latter coincides with the bidirectional migration of Javanese workers between Indonesia and Surinam and subsequent immigration to the Netherlands.

Therefore, it is not surprised that the TMRA of some HCV subtype (a1) dated back to 200 years ago. 

These HCV sequences contain information about the rate of sequence evolution and consequently such data sets can be used to directly infer molecular phylogenies on a natural time-scale of months, years, or millennia.

In the past two decades, Bayesian method has been so popular that to reconstruct the epidemic history of RNA viruses, such as HIV, HCV, Ebola, and Zika, using it have been common things. Some of them with high quality have been published in Science and Nature.

In the revision, we included more description and reference about this method in our manuscript.

Line 359: What is the link between HIV, HCV, and Yunnan. It seems plausible that HCV could be spread by traditional tattooing, but this seems unlikely for HIV. Is HIV thought to have originated in China in this area? Was this conclusion arrive at through phylogenetic analysis? Again, more thorough explanation would be useful.

Response: It was thought that subtype B and CRF07_BC have originated Yunnan province. Yes, this conclusion was concluded through phylogentic and molecular clock analysis. HCV and HIV share routes of transmission and many people with HIV are co-infected with HCV, especially in people who inject drugs and former paid blood donors. Therefore, the history of the epidemic of the two viruses was intermixed.

The origin and evolutionary history of three main HIV subype (B, CRF01_AE, and CRF07_BC), which were responsible for approximately 85% infection in China, have been well characterized.

Li,et al. showed that subtype B epidemics among former blood donors and heterosexuals in inland China were most likely originated from a single founding subtype B strain that had been circulating among PWID in Yunnan province. Yunnan province plays a pivotal role in bridging the preexisting subtype B epidemics in south-east Asia with the subsequent epidemic among FPDs and heterosexuals in inland China. (AIDS,2012,26:877-84. )

Meng, et al. demonstrated that CRF07_BC was originated in 1993 in IDU in Yunnan province and then initially spread to Guangxi (eastern neighbor to Yunnan) in 1994, to Xinjiang (northwest) in 1995 and to Sichuan (northern neighbor to Yunnan) in 1996. (PLoS ONE 7(12): e52373. )

Feng, et al. identified seven distinct phylogenetic clusters of CRF01_AE in China. Molecular clock analysis indicated that all CRF01_AE clusters were introduced from Southeast Asia in the 1990s, coinciding with the peak of Thailand’s HIV epidemic and the initiation of China’s free overseas travel policy for their citizens, which started with Thailand as the first destination country.(AIDS 2013, 27:1793-1802.)

We included the above three literature to support the opinion that Yunnan was the early epicenter of HIV in China. 

Minor points:

- Please use person first language – e.g. people who inject drugs rather than IDU

Response: We used the person first language in the new revision. 

We appreciate for Editors/Reviewers’ warm work earnestly. We acknowledged that it was difficult to incorporated all comments, and we just hoped that the revision is acceptable. Once again, thank you very much for your comments and suggestions.

---

## [Decision Letter · Decision Letter 1]

2 Oct 2023

PONE-D-22-30314R1Distribution pattern, molecular transmission networks, and phylodynamic of hepatitis C virus in ChinaPLOS ONE

Dear Dr. Lu,

Thank you for submitting your manuscript to PLOS ONE. After careful consideration, we feel that it has merit but does not fully meet PLOS ONE’s publication criteria as it currently stands. Therefore, we invite you to submit a revised version of the manuscript that addresses the points raised during the review process. Please address the concerns raised by Reviewer #1 in the revised manuscript prior to its acceptance.

We look forward to receiving your revised manuscript.

Kind regards,

Jason T. Blackard, PhD

Academic Editor

PLOS ONE

Journal Requirements:

Additional Editor Comments:

Please address the concerns raised by Reviewer #1 in the revised manuscript prior to its acceptance.

Reviewers' comments:

Reviewer's Responses to Questions

**Comments to the Author**

1. If the authors have adequately addressed your comments raised in a previous round of review and you feel that this manuscript is now acceptable for publication, you may indicate that here to bypass the “Comments to the Author” section, enter your conflict of interest statement in the “Confidential to Editor” section, and submit your "Accept" recommendation.

Reviewer #1: (No Response)

Reviewer #2: All comments have been addressed

2. Is the manuscript technically sound, and do the data support the conclusions?

Reviewer #1: (No Response)

Reviewer #2: Yes

3. Has the statistical analysis been performed appropriately and rigorously? 

Reviewer #1: (No Response)

Reviewer #2: I Don't Know

4. Have the authors made all data underlying the findings in their manuscript fully available?

Reviewer #1: (No Response)

Reviewer #2: Yes

5. Is the manuscript presented in an intelligible fashion and written in standard English?

Reviewer #1: (No Response)

Reviewer #2: Yes

6. Review Comments to the Author

Reviewer #1: The authors were diligent in making revisions to their manuscript in response to reviews.

* "The sequences whose sampling year is incongruent with genetic divergence were excluded for Bayesian analysis."

I apologize for making myself clearer in my previous review - my intention was to recommend that the authors should use TempEst to evaluate each data set for evidence of a molecular clock, and then to *discard the entire alignment for a given HCV subtype* if it did not meet this criterion. Individually discarding a substantial number of sequences based on a clock-based criterion is dangerous, because one can reshape a data set that has no temporal signal into one that strongly supports a clock. Rather, I think this treatment is associated with identifying sequences with incorrect dates, or from external evidence of systematic and substantial sequencing errors, for example.

As a compromise:

1. Please provide your exact quantitative criteria for filtering sequences at this step.

2. I am somewhat concerned that the results were quite sensitive to filtering sequences, i.e., estimates of TMRCA. The most transparent thing to do here would be to present both sets of results: (1) the original results without filtering sequences, and (2) results having filtered sequences under clearly reported criteria. The original results can be moved to Supplementary Materials to minimize revisions to the main text.

* "For each dataset, three independent Markov chain Monte Carlo (MCMC) chains were run for 100 million generations with states sampled every 10,000 generations. Log files were combined using Logcombiner to ensure sufficient convergence [...]"

More miscommunication here. The comment in my previous review was: "The results would also be more convincing if the authors had run replicate chains for each dataset to demonstrate convergence." I should have clarified that the replicate chain samples should be compared for evidence of convergence, *e.g.* overlapping posterior traces, before combining samples. It does seem that the authors did something like this, however, given the following line: "The convergence of MCMC chains was checked using Tracer (version 1.7.2)" If this is correct, then please provide the specific criteria you used to determine convergence - even visual assessment is okay. (On reflection, I suppose that one might demonstrate convergence if combining chains increases ESS.)

* the Open Science Framework URL does not work as written. Please use https://doi.org/10.17605/OSF.IO/NKD8Y

AP

Reviewer #2: The authors' revisions are appreciated and I feel they have sufficiently addressed my questions and comments.

7. PLOS authors have the option to publish the peer review history of their article (what does this mean?). If published, this will include your full peer review and any attached files.

Reviewer #1: **Yes: **Art Poon

Reviewer #2: No

---

## [Author Response · Author response to Decision Letter 1]

11 Nov 2023

PONE-D-22-30314R1

Distribution pattern, molecular transmission networks, and phylodynamic of hepatitis C virus in China

Dear Editor and Reviewers:

We are very grateful to you for giving us an opportunity to revise our manuscript. We really appreciate you very much for your positive and constructive comments and suggestions on our manuscript. We have studied Poon' comments carefully and tried our best to revise our manuscript according to the comments. The following are the responses and revisions we have made in response to the Poon' questions and suggestions point-by-point.

Thanks again to the hard work of the editor and reviewers.

Dear Dr. Lu,

Thank you for submitting your manuscript to PLOS ONE. After careful consideration, we feel that it has merit but does not fully meet PLOS ONE’s publication criteria as it currently stands. Therefore, we invite you to submit a revised version of the manuscript that addresses the points raised during the review process.

Please address the concerns raised by Reviewer #1 in the revised manuscript prior to its acceptance.

Reviewer #1: The authors were diligent in making revisions to their manuscript in response to reviews.

Response: Thank you for your positive comments. I learn a lot from your comments. Thanks.

* "The sequences whose sampling year is incongruent with genetic divergence were excluded for Bayesian analysis."

I apologize for making myself clearer in my previous review - my intention was to recommend that the authors should use TempEst to evaluate each data set for evidence of a molecular clock, and then to *discard the entire alignment for a given HCV subtype* if it did not meet this criterion. Individually discarding a substantial number of sequences based on a clock-based criterion is dangerous, because one can reshape a data set that has no temporal signal into one that strongly supports a clock. Rather, I think this treatment is associated with identifying sequences with incorrect dates, or from external evidence of systematic and substantial sequencing errors, for example.

As a compromise:

1.Please provide your exact quantitative criteria for filtering sequences at this step.

Response: 

I agree with your opinion concerning discarding a substantial number of sequences based on clock-based criterion. 

Our study invariably falls within the caveat of sampling bias.

We acknowledge this in the limitations section.

In HIV molecular epidemiology studies, researchers always need to discard some sequences either to reduce the computational load or to avoid potential bias that may arise from over-sampling a particular location.

In discarding sequences, we referred to the article by Andrew Rambaut,et al. entitled “Exploring the temporal structure of heterochronous sequences using TempEst (formerly Path-O-Gen).” published in Virus Evol 2016;2(1):vew007. 

The quantitative criterion for filtering sequences is their positions in the root-to-tip plot.

Outliers were carefully investigated and discarded if they were suspected of having quality problems.

In total, we discarded 152(6.5%) sequences from the original datasets which consist of 2342 sequences.

The problems for these outliers sequences were 1)low sequencing quality, 2)errors in sequence assembly, 3) alignment error in part of the sequence.

The majority of the 152(88.2%) sequences were retrieved from GenBank.

The TMRCA of the original datasets has a larger interval and lower ESS value, suggesting that including sequences with problems could bias the inference.

2. I am somewhat concerned that the results were quite sensitive to filtering sequences, i.e., estimates of TMRCA. The most transparent thing to do here would be to present both sets of results: (1) the original results without filtering sequences, and (2) results having filtered sequences under clearly reported criteria. The original results can be moved to Supplementary Materials to minimize revisions to the main text.

Response:

We provided the original results without filtering sequences in the Supplementary Materials.

* "For each dataset, three independent Markov chain Monte Carlo (MCMC) chains were run for 100 million generations with states sampled every 10,000 generations. Log files were combined using Logcombiner to ensure sufficient convergence [...]"

More miscommunication here. The comment in my previous review was: "The results would also be more convincing if the authors had run replicate chains for each dataset to demonstrate convergence." I should have clarified that the replicate chain samples should be compared for evidence of convergence, *e.g.* overlapping posterior traces, before combining samples. It does seem that the authors did something like this, however, given the following line: "The convergence of MCMC chains was checked using Tracer (version 1.7.2)" If this is correct, then please provide the specific criteria you used to determine convergence - even visual assessment is okay. (On reflection, I suppose that one might demonstrate convergence if combining chains increases ESS.)

Response:

The biggest difficulty we faced was the inclusion of too many datasets(8 NS5B+8 CE2+3 combination) in our analysis.

Moreover, we did not down-sample sequences to maintain the integrity of the original data.

Therefore, the computational load was large.

We ran multiple MCMC and combined the log files to increase ESS.

Fortunately, of the 19 datasets, only six have ESS value less than 200 in Bayesian inference.

We only had to run multiple for these six datasets.

Before we combine these results, we compare the evidence of convergence, e.g. ,statistics such as joint, prior, and likelihood.

I beg your pardon for the meaning of “overlapping posterior traces.”

We referred to at least nine classic papers on Bayesian evolutionary analysis, and none of them provided direct answer.

Would you kindly recommend some manuscripts concerning “overlapping posterior traces” .

Your kindness and help are greatly appreciated.

1.Nuno R Faria,et al.HIV epidemiology. The early spread and epidemic ignition of HIV-1 in human populations.Science 2014;346(6205):56-61. 

2.Claudia Palladino,et al.Epidemic history of hepatitis C virus genotypes and subtypes in Portugal.Sci Rep 2018;8(1):12266. 

3.Medhat K Shier ,et al.Molecular characterization and epidemic history of hepatitis C virus using core sequences of isolates from Central Province, Saudi Arabia.PLoS One

2017;12(9):e0184163.

4.Oliver G. Pybus,Eleanor Barnes,Rachel Taggart,et al.Genetic History of Hepatitis C Virus in East Asia.J Virol 2009; 83,1071-82.

5.Anna L McNaughton, Iain Dugald Cameron,Elizabeth B Wignall-Fleming,et al.2015. Spatiotemporal reconstruction of the introduction of hepatitis C virus into Scotland and its subsequent regional transmission. J Virol 2015; 89:11223–11232.

6.K Hoshino ,et al.Phylogenetic and phylodynamic analyses of hepatitis C virus subtype 1a in Okinawa, Japan.J Viral Hepat 2018;25(8):976-985. 

7.S Zhou,E Cella,W Zhou,et al. Population dynamics of hepatitis C virus subtypes in injecting drug users on methadone maintenance treatment in China associated with economic and health reform. J Viral Hepat 2017; 551-560.

8.Anna L McNaughton, Iain Dugald Cameron,Elizabeth B Wignall-Fleming,et al.2015. Spatiotemporal reconstruction of the introduction of hepatitis C virus into Scotland and its subsequent regional transmission. J Virol 2015; 89:11223–11232.

9.Peter V Markov,et al.Colonial history and contemporary transmission shape the genetic diversity of hepatitis C virus genotype 2 in Amsterdam.J Virol 2012 ;86(14):7677-87.

* the Open Science Framework URL does not work as written. Please use https://doi.org/10.17605/OSF.IO/NKD8Y.

Response: We used it.

We acknowledge that we have not encompass all of your comments.

We hope that these responses are acceptable.

Thank you very much.

 Yours sincerely

 Hongyan Lu.

---

## [Decision Letter · Decision Letter 2]

6 Dec 2023

Distribution pattern, molecular transmission networks, and phylodynamic of hepatitis C virus in China

PONE-D-22-30314R2

Dear Dr. Lu,

We’re pleased to inform you that your manuscript has been judged scientifically suitable for publication and will be formally accepted for publication once it meets all outstanding technical requirements.

Kind regards,

Jason T. Blackard, PhD

Academic Editor

PLOS ONE

Additional Editor Comments (optional):

None

Reviewers' comments:

Reviewer's Responses to Questions

**Comments to the Author**

1. If the authors have adequately addressed your comments raised in a previous round of review and you feel that this manuscript is now acceptable for publication, you may indicate that here to bypass the “Comments to the Author” section, enter your conflict of interest statement in the “Confidential to Editor” section, and submit your "Accept" recommendation.

Reviewer #1: All comments have been addressed

2. Is the manuscript technically sound, and do the data support the conclusions?

Reviewer #1: Yes

3. Has the statistical analysis been performed appropriately and rigorously? 

Reviewer #1: Yes

4. Have the authors made all data underlying the findings in their manuscript fully available?

Reviewer #1: Yes

5. Is the manuscript presented in an intelligible fashion and written in standard English?

Reviewer #1: Yes

6. Review Comments to the Author

Reviewer #1: I am pleased to recommend the revised manuscript for acceptance.

Regarding the authors' question about "overlapping posterior traces", this assessment is often visual (opening the replicate log files in Tracer, selecting all posterior traces in a combined plot and visually inspecting the resulting combined plots to determine whether the traces overlap). A more reproducible approach would be to use a convergence diagnostic like the Gelman-Rubin convergence diagnostic, which is available in the R package coda. There is no need to change anything in your manuscript, I am just trying to answer your question.

AP

7. PLOS authors have the option to publish the peer review history of their article (what does this mean?). If published, this will include your full peer review and any attached files.

Reviewer #1: **Yes: **Art Poon

---

## [Editor Report · Acceptance letter]

12 Dec 2023

PONE-D-22-30314R2 

PLOS ONE

Dear Dr. Lu, 

I'm pleased to inform you that your manuscript has been deemed suitable for publication in PLOS ONE. Congratulations! Your manuscript is now being handed over to our production team.

Kind regards, 

on behalf of

Dr. Jason T. Blackard 

Academic Editor

PLOS ONE